# Exploring the Perspectives of Canadian Clinicians Regarding Digitally Delivered Psychotherapies Utilized for Trauma-Affected Populations

**DOI:** 10.3390/ijerph22010081

**Published:** 2025-01-09

**Authors:** Sidney Yap, Rashell R. Allen, Katherine S. Bright, Matthew R. G. Brown, Lisa Burback, Jake Hayward, Olga Winkler, Kristopher Wells, Chelsea Jones, Phillip R. Sevigny, Megan McElheran, Keith Zukiwski, Andrew J. Greenshaw, Suzette Brémault-Phillips

**Affiliations:** 1Department of Psychiatry, Faculty of Medicine and Dentistry, University of Alberta, Edmonton, AB T6G 2H5, Canada; syap@ualberta.ca (S.Y.); burback@ualberta.ca (L.B.); ow@ualberta.ca (O.W.); agreensh@ualberta.ca (A.J.G.); 2Heroes in Mind, Advocacy, and Research Consortium, Faculty of Rehabilitation Medicine, University of Alberta, Edmonton, AB T6G 2G4, Canada; kbright@ualberta.ca (K.S.B.); mbrown2@ualberta.ca (M.R.G.B.); cweiman@ualberta.ca (C.J.); psevigny@ualberta.ca (P.R.S.); 3School of Clinical Child Psychology, Faculty of Education, University of Alberta, Edmonton, AB T6G 2G5, Canada; wozniak@ualberta.ca; 4School of Nursing and Midwifery, Mount Royal University, Calgary, AB T3E 6K6, Canada; 5Department of Occupational Therapy, Faculty of Rehabilitation Medicine, University of Alberta, Edmonton, AB T6G 2G4, Canada; 6Department of Computing Science, Faculty of Science, University of Alberta, Edmonton, AB T6G 2E8, Canada; 7Department of Emergency Medicine, Faculty of Medicine and Dentistry, University of Alberta, Edmonton, AB T6G 2T4, Canada; jhayward@ualberta.ca; 8Department of Child and Youth Care, Faculty of Health and Community Studies, MacEwan University, Edmonton, AB T5J 4S2, Canada; kristopher.wells@macewan.ca; 9Faculty of Education, University of Alberta, Edmonton, AB T6G 2G5, Canada; 10Wayfound Mental Health Group, Calgary, AB T2R 1J5, Canada; meganm@wayfound.ca; 11Haikei Health, Edmonton, AB T5W 3H1, Canada; keith@drzukiwski.com

**Keywords:** web-based intervention, psychotherapy, access to therapies, trauma, trauma-focused psychotherapy, health services, military, veteran, public safety personnel, implementation science

## Abstract

Many clinical sites shifted towards digital delivery of mental health services during the COVID-19 pandemic. There is still much to learn regarding tailoring digitally delivered interventions for trauma-affected populations. The current study examined the perceptions of Canadian mental health clinicians who provided digitally delivered psychotherapies utilized for trauma-affected populations. Specifically, we explored the shift to digital health use, what changed with this rapid shift, what needs, problems, and solutions arose, and important future considerations associated with delivering trauma-focused and adjunct treatments digitally. Survey data were collected from 12 Canadian mental health clinician participants. Surveys were adapted from the Alberta Quality Matrix of Health and Unified Theory of Acceptance and Use of Technology model. As a follow-up, the participants were invited to participate in either a semi-structured qualitative interview or focus group to further explore their perspectives on digitally delivered trauma-focused and adjunct therapies. Twenty-four clinician participants partook in an interview or focus group. The participants in this study supported the use of digitally delivered psychotherapies utilized for trauma-affected populations, sharing that these interventions appeared to offer similar quality of care to in-person delivery. Further research is required to address clinicians’ concerns with digital delivery (e.g., patient safety) and identify other avenues in which digitally delivered psychotherapies utilized for trauma-affected populations can be engaged with and improved upon.

## 1. Introduction

An escalation of mental health concerns stemming from the onset of COVID-19 and the need to adhere to COVID-19-related physical distancing mandates and government restrictions prompted rapid changes in mental healthcare service delivery. The pandemic required an unprecedented rapid transition from in-person to digital methods (e.g., teletherapy, telemedicine, eHealth, mobile health) [1]. Considering the mental health consequences of the COVID-19 pandemic [2,3], digital mental health interventions (DMHI) offered a cost-effective alternative to in-person mental health treatment while complying with pandemic-related public health requirements for physical distancing [4]. Such interventions allowed for access to timely and secure trauma therapies that are critical to supporting the transdiagnostic mental health needs of public safety personnel (PSP), military members, and Veterans [5], many of whom have experienced potentially psychologically traumatic event exposures (PPTEs, e.g., exposure to actual or threatened death, serious injury, or sexual violence) [6,7]. Accordingly, the COVID-19 pandemic greatly accelerated the uptake of DMHIs [8].

Research indicates that digitally delivered prolonged exposure (PE), eye movement desensitization and reprocessing (EMDR), and trauma-focused cognitive behavioral therapy (TF-CBT) may be effective at reducing PTSD symptomatology [9,10]. Digital mental health interventions generally provide clients with increased convenience, comfort, and access to treatment [11,12]. Further, increased access to care can lead to substantial cost and time savings for clients [4]. Finally, digital delivery may aid in decreasing the stigma related to accessing mental healthcare services [13].

Clinician experience and comfort with delivering DMHI can appreciably impact treatment success. The majority of mental health clinicians did not have experience providing digital services, trauma-specific or otherwise, prior to the onset of the COVID-19 pandemic [14,15]. Clinicians have reported many concerns with providing DMHI, including a lack of emotional connection with patients, increased distraction (for both clients and clinicians), concerns with privacy and confidentiality, and clinician work–life balance [14]. Maintaining client engagement during digitally delivered group-based treatments is an additional challenge, with therapists perceiving online psychotherapy groups to be less effective [16]. Many also cite a lack of specific training for providing online group therapy as a barrier to providing such interventions [16]. Much of the extant literature has also primarily focused on DMHI generally, such as the use of Internet-based cognitive behavioral therapy in civilian populations [17,18], with little focus on how the shift to digital delivery affected PPTE-focused treatments specifically. These challenges may contribute to the low acceptability of DMHI among clinicians [19], even with the knowledge that DMHI may provide many unique benefits.

The aim of this study was to identify the strengths and weaknesses of digitally delivered psychotherapies utilized for trauma-affected populations based on the perceptions of Canadian mental health clinicians who have provided these interventions. Surveys adapted from the Alberta Quality Matrix for Health (AQMH) [20] and the Unified Theory of Acceptance and Use of Technology (UTAUT) [21] and semi-structured interviews and focus groups were utilized to collect data. The state of implementation of DMHIs from the perspectives of clients, clinicians, and community partners was reported on previously [22].

## 2. Methods

The current study, in a community-engaged research setting [23], used a mixed-methods design [24]. The mental health clinician participants completed a demographic survey, and surveys adapted from the AQMH and UTAUT. All surveys were administered through Research Electronic Data Capture (REDCap 14.5.2), a secure web application used for building, managing, and completing online surveys [25,26,27]. The participants were also invited to partake in a 30–60 min semi-structured interview or focus group, conducted over Zoom (version 5.15). These interviews were conducted to gain insights into the participants’ experiences of providing digitally delivered psychotherapies utilized for trauma-affected populations. Data were stored on the REDCap server and a dedicated, encrypted, and password-protected research drive hosted by the Faculty of Rehabilitation Medicine at the University of Alberta. These data were only accessible to research team members.

Ethics approval was obtained from the University of Alberta’s Health Research Ethics Board (Pro00109065) prior to the commencement of research activities. Written informed consent was obtained from all study participants prior to commencing research activities.

### 2.1. Participant Inclusion and Exclusion Criteria

Multidisciplinary mental health clinicians and providers who practice in Canada and have provided digitally delivered psychotherapies utilized for trauma-affected populations to Canadian military members, Veterans, and PSP were recruited. This included, but was not limited to, psychiatrists, psychologists, social workers, and nurse practitioners who were eligible to provide psychotherapy in Canada in accordance with their provincial regulatory bodies.

Mental health clinicians who were not practicing in Canada, not delivering digital trauma therapies, and/or not working with Canadian military members, Veterans, and PSP were excluded from study participation. Individuals who were unable to provide informed written consent and/or were not fluent in English were also excluded.

The same inclusion and exclusion criteria were used in a previous publication [22].

### 2.2. Recruitment and Data Collection

Snowball and purposeful sampling strategies were used to recruit participants from partner and non-partner mental health clinics. Clinics were provided with recruitment materials and information, which were given to potential clinician participants. Interested individuals who completed a consent to contact form over REDCap were contacted by a member of the research team by telephone or email to discuss the study information, determine eligibility, and assess their willingness to voluntarily participate. A link to a REDCap webpage was shared with eligible individuals to access and complete an informed consent form. Recruitment took place between January 2022 and March 2023.

Participants who provided informed consent completed the survey measures and/or a semi-structured interview or focus group. Surveys were iteratively developed based on the AQMH [20] and UTAUT [21] by the research team. These measures were crafted to maximally align the survey language with each AQMH and UTAUT dimension, integrating principles of equity, diversity, and inclusion, minimizing survey burden, and maximizing data quality, collection, and analyses. Data collection took place from February 2022 to May 2023.

### 2.3. Tools and Measures

#### 2.3.1. Survey Based on Alberta Quality Matrix for Health

The AQMH was designed based on the work of *Crossing the Quality Chasm: A New Health System for the 21st Century* [28]. The AQMH is made up of two components: (1) dimensions of quality, which focuses on aspects of the patient and client experience, and (2) areas of need, which divides services provided by the health system into four distinct but related categories (being healthy, getting better, living with illness or disability, and end of life). The components are considered across the following six dimensions: acceptability, accessibility, appropriateness, effectiveness, efficiency, and safety [20].

The research team iteratively developed a survey based on the AQMH and previous literature (see Appendix A for a copy of the AQMH and the survey adapted from the AQMH). This survey consisted of 10 questions, scored on a 7-point Likert-type scale from 1 (*strongly disagree*) to 7 (*strongly agree*). This conversion allowed the participants to rate the quality of service of digital and in-person delivery of trauma therapies along the AQMH dimensions, using one question to assess each of the following ten criteria: ease of use, convenience, acceptability, practicality, accessibility, appropriateness, effectiveness, efficiency, safety, and fit.

#### 2.3.2. Survey Based on Unified Theory of Acceptance and Use of Technology

The UTAUT model was developed by Venkatesh et al. in 2003 as a synthesis of eight technology acceptance models. The UTAUT was designed to assess the acceptance of new technology and may explain up to 70% of the variance in intention to use technologies [21]. The UTAUT has well-established construct and content validity. The six factors influencing technology use as measured by the UTAUT include the following:Effort Expectancy: the degree of ease associated with using the technology. If participants perceive psychotherapies utilized for trauma-affected populations to have low Effort Expectancy, it would be expected that they would be more likely to use the technology.Performance Expectancy: assesses whether the participant believes that the technology would improve the performance of the job they are trying to complete. If belief in psychotherapies utilized for trauma-affected populations is high, participants would be more likely to use the technology.Behavioral Intention: the degree to which participants have a conscious plan to utilize technology. This construct in turn predicts Use Behavior and technology acceptance.Social Influence: the extent to which individuals surrounding the participant perceive the usefulness of psychotherapies utilized for trauma-affected populations and how much these important others’ influence participants’ use of psychotherapies utilized for trauma-affected populations.Facilitating Conditions: the extent to which conditions, such as organizational and technical infrastructure, surrounding the participant support the use of psychotherapies utilized for trauma-affected populations.Use Behavior: the extent to which participants use psychotherapies utilized for trauma-affected populations.

The research team iteratively developed this survey based on the UTAUT content and components, as well as the previous literature (see Appendix A for a copy of the UTAUT and the surveys adapted from the UTAUT). The survey based on the UTAUT consisted of 18 questions, scored on a 7-point Likert-type scale from 1 (*strongly disagree*) to 7 (*strongly agree*). Each of the 6 UTAUT constructs was measured individually, with 3 questions asked per construct. All construct scores were then combined to assess the overall useability of the technology employed for digitally delivered trauma therapy [21].

#### 2.3.3. Interviews and Focus Groups

Qualitative data were collected via 30–60 min semi-structured solo interviews (*n* = 7) or focus groups (*n* = 2) conducted and recorded over Zoom [29]. Semi-structured interviews and focus groups were conducted to identify the strengths and weaknesses and further explore participant perspectives of digitally delivered psychotherapies utilized for trauma-affected populations. Key topics of discussion included the previous and current state of implementation of digitally delivered psychotherapies utilized for trauma-affected populations during the COVID-19 pandemic; barriers to, facilitators of, and recommendations for the use of digitally delivered TFP; and the clinical effectiveness of digitally delivered interventions. See Appendix A for copies of the interview scripts used for clinician interviews and focus groups.

Focus groups were purposely heterogeneous with respect to professional representation and experience with using psychotherapies utilized for trauma-affected populations, including digitally delivered interventions. This design allowed for broad crosstalk and pollination of complementary and alternative ideas, experiences, and conversation, resulting in rich comprehensive data such that sufficient information power was reached [30].

### 2.4. Data Analysis

All data were de-identified prior to analysis. De-identified survey data were analyzed using IBM SPSS Statistics software (Version 28.0) [31]. NVIVO 13 (2020, R1) was used to facilitate qualitative analysis of the de-identified interview and focus group data.

Descriptive statistics were calculated for each survey variable for the clinician participants. Non-parametric analyses were conducted to account for the relatively limited sample size of the study. Paired-sample Wilcoxon signed-rank tests were used to assess within-subject differences (*p* ≤ 0.05) in the median AQMH survey dimension scores between digitally delivered and in-person psychotherapies. One-sample Wilcoxon signed-rank tests were used to assess within-subject differences (*p* ≤ 0.05) between the observed and reference median UTAUT survey dimension scores. For this analysis, the reference score was set to 12 (i.e., neutral score, based on the sum of three total questions per UTAUT dimension asked on a Likert scale from 1 to 7, where 4 is the median score for each question). In total, 10 and 6 tests were conducted for the AQMH and UTAUT survey results, respectively. To address multiple comparisons across the 16 statistical tests, the Benjamini–Hochberg procedure was used to control the false discovery rate (FDR) [32]. A summary of the statistical tests can be found in Appendix A.

Video-recorded interviews and focus groups were transcribed using Adobe Premiere Pro and thematically analyzed both deductively and inductively following an iterative process [33]. A hybrid coding approach was taken to analyze qualitative data. Deductively, initial codes were developed based on interview and focus group topics and study objectives. Inductive coding involved identifying themes that emerged from the collected data. Coding for each interview and focus group was independently conducted by two research team members (S.Y., R.W.). Following this, a senior researcher (S.B.-P.) reviewed and refined the codes. They were then combined and tabulated into preliminary themes. Analysis of preliminary themes by the larger research team followed, with differences being resolved through discussion. A proposed thematic theory then underwent collective analysis, where preliminary themes were modified and key quotes isolated to illustrate the selected themes. The final thematic narrative was then prepared. The Standard for Reporting Qualitative Research was used to guide the reporting process [34].

## 3. Results

### 3.1. Mental Health Clinician Demographics

Twelve Canadian mental health clinicians completed the survey measures. The clinician participants self-identified as female (*n* = 9), male (*n* = 3), women (*n* = 9), or men (*n* = 3), all as Caucasian, with an average age of 46 ± 8.3 years (range: 33 to 58 years). The participants had an average of 14 ± 6.3 years (range: 5 to 25 years) of clinical experience and 9 ± 5.3 years (range: 3 to 16 years) of experience specifically providing trauma therapies. Several participating clinicians (*n* = 5) had not used any form of digital delivery prior to the COVID-19 pandemic. During study participation, six clinicians exclusively provided digitally delivered care, while the remaining six clinicians provided a combination of digitally delivered and in-person care. Table 1 provides further demographic information.

Twenty-four clinician participants took part in a semi-structured interview or focus group. A total of 3 clinicians participated in a semi-structured interview (2 female, 1 male/2 women, 1 man; 3 Caucasian), while the remaining 21 participated in one of two focus groups (focus group 1: 19 females, 2 males/19 women, 2 men; focus group 2: 12 females, 2 males/12 women, 2 men). Participant workplaces included an occupational stress and injury clinic (*n* = 12), a provincial health authority (*n* = 10), or a private practice (*n* = 2). The clinicians had an average of 13 ± 9.7 years (range: 0 to 40 years) of clinical experience and 6 ± 4.4 years (range: 0 to 15 years) of experience specifically providing trauma therapies.

### 3.2. Survey Results

#### 3.2.1. Results of Survey Based on Alberta Quality Matrix for Health

There were no statistically significant differences between the clinician participant AQMH scores for digitally delivered vs. in-person therapy for all 10 AQMH dimensions: ease of use, convenience, acceptability, accessibility, practicality, appropriateness, effectiveness, efficiency, safety, and fit (Figure 1, Table 2).

#### 3.2.2. Results of Survey Based on Unified Theory of Acceptance and Use of Technology

Overall, the clinician participants strongly agreed that digitally delivered psychotherapies utilized for trauma-affected populations were a viable option when compared to in-person trauma therapies. The clinician participants indicated slight agreement, agreement, or strong agreement with the Effort Expectancy (18/21), Performance Expectancy (18.5/21), Behavioral Intention (21/21), Use Behavior (18.5/21), and Social Influence (16/21) constructs. Analysis revealed that the Behavioral Intention (*p* = 0.002), Performance Expectancy (*p* = 0.003), Use Behavior (*p* = 0.004), and Effort Expectancy (*p* = 0.012) scores were significantly different compared to the expected median score of 4 (on a Likert scale from 1 to 7), surviving FDR multiple comparison correction. The clinician participants indicated that they neither agreed nor disagreed with the Facilitating Conditions (15/21) construct (Figure 2, Table 3).

### 3.3. Interview Results

Thematic analysis of the interview data isolated four main themes regarding digitally delivered psychotherapies utilized for trauma-affected populations: (1) similarities between digitally delivered and in-person care; (2) unique benefits of digital delivery; (3) concerns with digital delivery; and (4) future directions and recommendations.

Theme 1: Similarities between digitally delivered and in-person care.

The clinician participants shared that digitally delivered psychotherapies utilized for trauma-affected populations provided similar quality of care to in-person delivery. They shared that patient assessments and certain treatment modalities, including EMDR and CBT, were easy to adapt to the digital environment and resulted in successful administration of treatment with little to no discernable differences to in-person delivery.


*“I’ve had people who’ve done full courses of treatment doing EMDR online, I’ve actually never met them in person. It was beautiful, very effective”.*

*[Clinician participant 3]*


Theme 2: Unique benefits of digital delivery.

The clinician participants believed the most crucial benefit of digital delivery was that it increased the accessibility of psychotherapies utilized for trauma-affected populations. Having digital therapy sessions allowed for more flexible scheduling and allowed clinicians to reach patients living in remote locations. Prior to the rise of digitally delivered psychotherapy, such clients would typically have to spend large amounts of resources to travel to urban centers to receive treatment. If clients did not have access to these resources, they would not be able to receive treatment.


*“[A]s you know we work with clients in the Northwest Territories, and it’s kind of revolutionized care for them, because going through the old system they had to go [into] a mental health clinic. It was, it was really hard for them because they were in small communities[…] The other thing is that for people who are far away, before they couldn’t take part in groups, cause it took too long to get here, now they can join any group they want”.*

*[Clinician participant 4]*


Some clinician participants revealed that digital delivery aided in their therapeutic duties in creative ways not thought of when delivering in-person care. This allowed a clinician the opportunity to conduct more comprehensive assessments of their clients. For example, working digitally allowed a clinician participant to see into their clients’ homes, allowing for a better understanding of their situation and clarifying the next treatment steps.


*“[W]orking virtually actually allowed me [to make] a better assessment of what was going on. So, you know, I had one client who talked a lot about how messy his house is and how he can’t keep up with household tasks, and he’s just kinda failing, he has zero motivation to do the dishes. And so I actually had him bring me over and show me the dishes[…] working virtually allowed me to do CBT in action rather than kind of talking about it theoretically. So rather than talking about you know how starting you know avoiding, avoidance with doing kind of small tasks improved mood, and I did this with more than one client, you know they would talk about this thing they are avoiding, you know I would have them take a step”.*

*[Clinician participant 3]*


For another clinician, digital delivery provided them with a canvas to seek out unique ways of making the therapeutic space more comfortable for their clients, which would not have been done while providing in-person care.


*“And it helps, most of them actually [like] to use sounds. And in my office, interestingly the sound is just beep, beep, beep, beep, back and forth. And on the computer there’s 30 sounds I can choose from. And so one guy[,] mainly for the calm place, I used a very simple sort of wood chopping sound during processing. But if we go to a calm place at the end of the session, one guy his calm place is a BC River Canyon where there’s a river. So we put the river back and forth and then another guy likes the beach so we put the beach back and forth and he [hears] the sound of the wave. So that’s kind of neat to be able to do that”.*

*[Clinician participant 18]*


Theme 3: Concerns with digital delivery.

The clinician participants indicated that a major barrier to providing digitally delivered psychotherapies for trauma-affected populations was the lack of preparedness stemming from the sudden shift to using digital platforms following the onset of the COVID-19 pandemic. Clinicians shared that they received little to no support to prepare for delivering digital psychotherapies for trauma-affected populations and were tasked with troubleshooting technical issues with very limited support, leading to frustration and discomfort.


*“[A gap that should be addressed is providing] practical training in using Zoom and troubleshooting issues for clinicians who aren’t as tech savvy”.*

*[Clinician participant 8]*


These challenges greatly impacted many clinicians’ therapeutic ability, disrupting the flow of treatment and leaving them feeling like they were losing control over the therapeutic environment.


*“I could add that I have significantly less control, if not no control, over the confidential therapeutic space, at least on the end of the client in virtual therapy”.*

*[Clinician participant 2]*


The clinician participants also expressed concern regarding how differing levels of technological literacy could affect clients, leading some clinicians to wonder if digital delivery truly aids in creating more equitable access to care.


*“Virtual care operates on the assumption clients have access to technology to support this, but not all clients may have the funds and or literacy to access this technology”.*

*[Clinician participant 13]*


Further, the clinician participants shared concerns about client safety. The clinician participants were unsure of how to respond to dangerous relationships their clients were a part of or adverse events their clients may experience while attending digitally delivered psychotherapy sessions.


*“If a client is in a relationship that has violence or has a history of domestic violence[,] that has been something that’s come up for me[,] and I’ve had to kind of check in with the client every time, is your partner home, are they able to hear it hear in and listen in on this, and I was able to sort of [schedule] with the client around having sessions when the partner was not home”.*

*[Clinician participant 2]*



*“[There] sort of [has to be] concerns for safety. How do I help my client that’s having an over-the-top reaction or different reactions if there’s something[?] [Y]ou’re not there to deal with it directly, you know, especially if they cut the mic or whatever, it can [happen] abruptly. Then you’re like “what the hell is happening there?” I can’t be there. No. So yeah, it doesn’t come up very often, hardly at all. But, you know, there’s something to be concerned about”.*

*[Clinician participant 18]*


Finally, the clinician participants worried that digitally delivered psychotherapies utilized for trauma-affected populations would not provide effective treatment for certain clients. Specifically, they feared that clients with highly avoidant behavior, dissociative tendencies, complex trauma, or emotional dysregulation may not attain the same benefits from digitally delivered trauma therapy as in-person therapy. For example, some clients who were using digitally delivered psychotherapy exhibited an emergence of avoidant behaviors following treatment commencement. As avoidance (implicit and explicit) may contribute to treatment resistance/poor treatment response, these concerns must be addressed moving forward.


*“I’ve had a few people who are a combination of really avoidant, so [strong] avoidance component of PTSD, and then also tend to have maybe some [obsessive compulsive] personality traits, where they’re a combination of [symptoms]. If we’re doing CPT for example, there’s kind of a lot of really subtle avoidance, and often sort of getting into rumination in the session rather than focusing on their [stuck point][…]”*
[*Clinician participant 7]*


*“[W]hat I’m finding is that [clients are] starting to isolate more at home, and not engaging in their community or getting out, and everything has defaulted back to the caregivers in the home, it’s that trust. So I really noticed that in a couple of clients who were already highly anxious and highly avoidant. So I don’t think it [the] virtual serves a good purpose in that sense”.*

*[Clinician participant 9]*


Theme 4: Future directions and recommendations.

The clinician participants believed that digital delivery would continue to be offered moving forward and encouraged the wider spread of digital delivery such that more clients would have the opportunity to receive care.


*“Perhaps increasing [digital delivery or hybrid care] outside of the Edmonton zone. […] So if people are interested in these groups, having it more accessible. We got better in terms of the handouts, having it all prepackaged, everything ready to go before the group because initially we were going between all the sessions, session one, session two, and so forth. So I think we’re more organized, but improving it [and] having more flexibility is important”.*

*[Clinician participant 17]*


Clinicians recommended that certain changes would aid in increasing treatment accessibility, use, and retention. They believed that hybrid care, a combination of digital and in-person psychotherapeutic services, would offer a balance between providing the care clients are seeking and the delivery method they prefer.


*“Perhaps like increasing [digital delivery or hybrid care] […] to provide clients with the care they are seeking”.*

*[Clinician participant 17]*


Finally, one clinician participant encouraged other clinicians to be open about providing digital care and integrating digital delivery into their practice.


*“And I think the other thing is just to, you know, embrace [digital delivery] [and] recognize that there are some really significant benefits to it. You know, try it out and see because I know some people are just really reluctant to even try it, you know, some people never [used] online delivery through [the pandemic]”.*

*[Clinician participant 19]*


## 4. Discussion

The current study provides preliminary evidence illustrating that Canadian mental health clinicians support the use of DMHI in the context of providing psychotherapy to military members, Veterans, and PSP who have experienced trauma. The clinician participants reported that digitally delivered psychotherapies utilized for trauma-affected populations appear to offer similar quality of care to in-person delivery. Challenges relating to providing digital care were identified, which must be addressed if the use of digital delivery is going to be expanded moving forward.

Canadian mental health clinicians face several unique challenges when providing care. For example, they must contend with the highly heterogeneous population of Canada. Canadian clinicians routinely provide care to clients that live in varied geographic locations (e.g., urban vs. rural areas). Their client base may be made up of a rich and diverse cultural background (e.g., white, Hispanic, Indigenous), such that the care they provide must be acceptable and appropriate for those from a variety of cultural backgrounds. Capturing the unique perspectives of Canadian mental health clinicians was a major priority for the research team. Although the sample size for this study was smaller than expected, the insights shared by the participants add valuable perspectives to the extant literature and highlight the importance of co-designing healthcare services with providers. Further, this study, along with a previously published report [22], provides a needed update on previous research in the field [5]. Continuing to conduct research in this rapidly evolving field will allow us to capture shifts in attitudes and the usage of digitally delivered psychotherapies utilized for trauma-affected populations as we move past the COVID-19 pandemic. Our findings may play an important role when considering the expansion of DMHI services within the general healthcare system and in delivering care to individuals living in rural or remote communities.

Clinician participant responses to the survey based on the AQMH indicated no significant differences in service quality between digitally delivered and in-person trauma therapies. This is consistent with previous research suggesting that equivalent quality of care is attainable via digital or in-person service delivery modalities [9,10]. The clinician interview data supported the findings of the survey based on the AQMH, sharing cases where EMDR and CBT were successfully adapted for digital delivery and treatment was administered with little to no discernable differences to in-person delivery. Clinicians also shared many key advantages of digital delivery through their interviews and focus groups, including increased accessibility of treatment, expanded clinical opportunities, and the creation of more comfortable therapeutic spaces for their clients. Further research is needed to clarify how digital delivery impacts the AQMH dimensions of healthcare and identify which dimensions need to be prioritized within healthcare programs to ensure the highest quality of care.

Clinician participant responses to the survey based on the UTAUT indicated that digitally delivered psychotherapies utilized for trauma-affected populations were highly usable, reflected by the relatively high scores for the Effort Expectancy, Performance Expectancy, Behavioral Intention, and Use Behavior constructs. The Social Influence and Facilitating Conditions constructs had relatively lower scores, indicating that they were not as prioritized by the clinician participants. Taken together, these results suggest that the clinician participants were providing digitally delivered psychotherapies despite a lack of perceived support from important figures in their lives, such as the organizations they work for (e.g., mental health clinics). The large range and standard deviations for the Facilitating Conditions construct were noteworthy, however, as they potentially indicate that this construct is highly dependent on the context (e.g., workplace) and clinician (e.g., individual need for support). Factors related to Facilitating Conditions, such as organizational support, therefore require further study as it is unclear based on our results whether sufficient support is being provided to clinicians to maintain the long-term viability of digitally delivered psychotherapies. Finally, although there are moderators known to influence Behavioral Intention and overall technology acceptance, the influence of these moderators was not evaluated due to the limited sample size. Similar issues were also reported in a report regarding client perspectives on the usability of digitally delivered psychotherapies utilized for trauma-affected populations [35].

The clinician participants raised multiple concerns regarding the support provided while providing digitally delivered psychotherapies utilized for trauma-affected populations during the interview and focus group sessions. In particular, the clinician participants expressed frustration stemming from a lack of training and easily accessible technological support. Similar findings were found in a qualitative study conducted in the United States, which recommended that, amongst other factors, providing adequate technology and support for mental health providers and local behavioral health departments would be necessary for the successful implementation of DMHI [36]. Further research is needed to better understand if these factors should be prioritized when implementing DMHI, including psychotherapies utilized for trauma-affected populations.

The lack of control over the therapeutic space while providing digitally delivered care was another concern shared by the clinician participants. Previous research posits that those who have experienced interpersonal trauma may have difficulties developing strong therapeutic alliances [37], which may affect a clinician’s perceived control over the psychotherapy session. Indeed, a tenuous relationship with their clients may lead clinicians to feel that they have less control over the therapeutic environment and increase their worry regarding clients experiencing distress when providing digital care. The clinicians shared that they were unsure of how to navigate such difficult situations, signaling that there is a need for the creation of clinical practice policies and guidelines regarding managing client crises when working digitally. Further, for the safety of mental health clinicians, guidelines clarifying the legal protections they have when providing DMHI should be developed. Finally, it would be useful to further evaluate whether there are differences in implementation depending on the context in which a clinician provides clinical services (e.g., is DMHI implementation more likely in a clinic-based setting versus an independent practice?).

### Limitations

There were several study limitations which must be acknowledged. First, all recruitment and surveys utilized in this study were in English, limiting responses from non-English speaking communities. Second, the age and experience of our clinician population may have biased our results. The clinician participants in this study were highly experienced and had an average of 14 years of clinical experience. It is possible that less experienced clinicians may have had a more negative experience providing DMHIs. Similar limitations have been described in previous research [16]. Finally, the sample size and the diversity of our study population were relatively limited, which precluded analyzing certain mediating effects. For example, the influence of moderators known to influence Behavioral Intention and overall technology acceptance were not evaluated due to our limited sample. Potential gender and sex differences were also not explored given the limited sample size.

Future studies with larger sample sizes would be useful to replicate the findings reported here and to explore potential moderating factors, such as differences related to cultural background, gender, sex, and sexual orientation with regard to the acceptance of digitally delivered care.

## 5. Conclusions

The mental health clinician participants shared common perspectives that demonstrated the unique benefits and disadvantages of providing digitally delivered psychotherapies utilized for trauma-affected populations. Given the high rate of PTSIs within this client base, it is critical that clinicians can provide high-quality interventions in a secure, cost-effective, and accessible manner. Our results suggest that digital delivery offers an accessible and practical way for Canadian military members, Veterans, and PSP to receive trauma therapy. Further, the clinician participants indicated that hybrid care, a mixture of digital and in-person delivery, should be expanded upon in the future to ensure client populations are receiving care through their preferred delivery method. The lives of trauma-affected populations may be directly impacted and improved with the use of DMHI.

## Figures and Tables

**Figure 1 ijerph-22-00081-f001:**
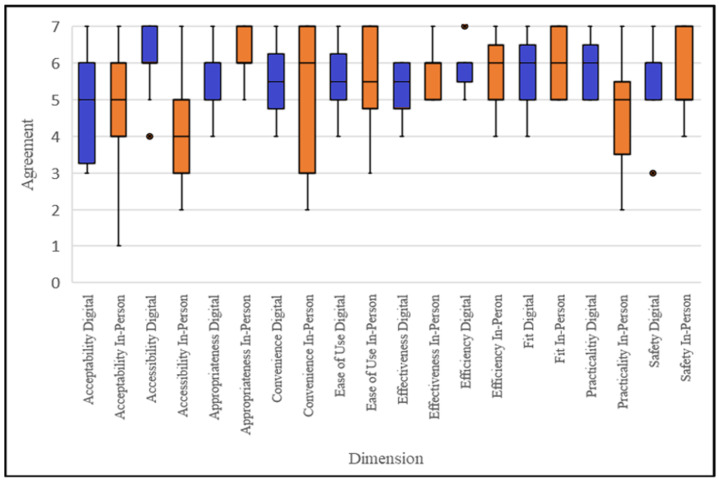
Box and whisker plots indicating clinician participant (*n* = 12) median AQMH survey scores, first and third quartiles, and minimum and maximum scores. Blue refers to digital delivery; orange refers to in-person delivery. ● indicates outlier.

**Figure 2 ijerph-22-00081-f002:**
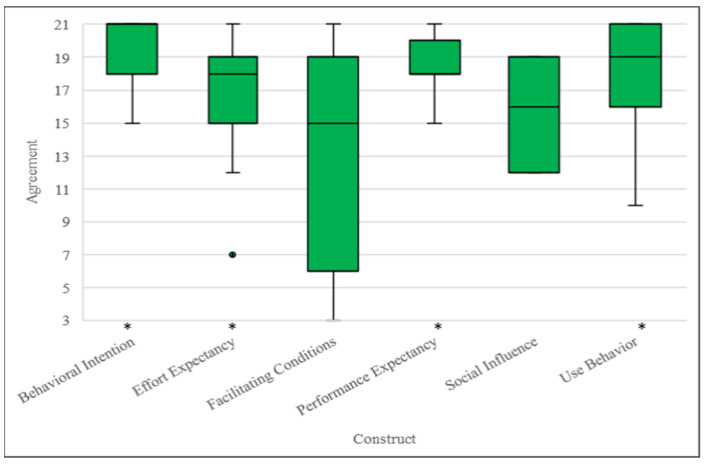
Box and whisker plots indicating clinician participant (*n* = 12) median UTAUT construct scores, first and third quartiles, and minimum and maximum scores. * = significant difference (*p* < 0.05) between median score and a reference score of 12 (total score of three questions asked based on a Likert scale from 1 to 7) based on one-sample Wilcoxon signed-rank test, corrected for multiple comparisons. ● indicates outlier.

**Table 1 ijerph-22-00081-t001:** Employment information for clinicians who completed survey measures.

Participant Clinical Role	Number of Participants
Registered Psychologist	4
Social Worker	4
Psychiatrist	2
Mental Health Therapist	2
Participant Workplace	Number of Participants
Occupational Stress and Injury Clinic	5
Provincial Health Authority	3
Private Practice	3
Regional Health Service	1
Therapeutic Modalities Provided Digitally ^1^	
Eye Movement Desensitization and Reprocessing	10
Cognitive Behavioral Therapy ^2^	7
Cognitive Processing Therapy	6
Dialectical Behavior Therapy	5
Prolonged Exposure	4
Mindfulness/Self-Compassion	3
Somatic Therapy	2
Acceptance and Commitment Therapy	2
Neurotherapy	1

^1^ Therapeutic modalities were used to treat symptoms related to diverse psychological traumas, including, but not limited to, operational/occupational injuries, adverse childhood events, and sexual abuse/rape. ^2^ Unknown if cognitive behavioral therapy interventions were specifically trauma-focused.

**Table 2 ijerph-22-00081-t002:** Clinician participant (*n* = 12) median AQMH survey statistical analysis results.

AQMH Dimension	*p*-Value
Ease of Use	0.943
Convenience	0.681
Acceptability	0.952
Practicality	0.114
Accessibility	0.020
Appropriateness	0.046
Effectiveness	0.194
Efficiency	0.785
Safety	0.492
Fit	0.395

**Table 3 ijerph-22-00081-t003:** Clinician participant (*n* = 12) UTAUT survey statistical analysis results. * = significant difference (*p* < 0.05) between median score and a reference score of 12 (total score of three questions asked based on a Likert scale from 1 to 7) based on one-sample Wilcoxon signed-rank test, corrected for multiple comparisons.

UTAUT Dimension	*p*-Value
Performance Expectancy	0.003 *
Effort Expectancy	0.012 *
Social Influence	0.017
Facilitating Conditions	0.593
Behavioral Intention	0.002 *
Use Behavior	0.004 *

## Data Availability

The original contributions presented in this study are included in the article; further inquiries can be directed to the corresponding author.

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
