# Peer review of "Exploring the Perspectives of Canadian Clinicians Regarding Digitally Delivered Psychotherapies Utilized for Trauma-Affected Populations"

_ijerph, 2025, doi:10.3390/ijerph22010081_

Round 1
Reviewer 1 Report
Comments and Suggestions for Authors
Thank you for the opportunity to review this is novel study exploring clinician perspectives in Alberta, Canada on the utility of virtual delivery of therapies to people who have experienced trauma .
The focus on the study is the mode of delivery, its acceptability according to specific tools used to measure healthcare quality and acceptability of digital tools. It is not, therefore, specifically focused on measuring the effectiveness of the therapy on patient progress.
The introduction and discussion are clear and methods well described. The increase in this mode of delivery of therapies is an important new area and the clinician perspectives of both risks and advantages are interesting and informative.
General comments:
There are several spelling and punctuation errors. Please see comments in the attached file.
The formatting of in-text citations and the bibliography needs attention.
Can you comment on any differences in the perceived utility and safety of this approach with trauma-affected populations compared to other mental health difficulties not specifically attributed to traumatic experiences?
Specific comments:
Line 55 please define PSP (public safety personnel)
Lines 131-137 could this be summarised in a table, and can you explain how you selected participants for interviews/focus groups?
Line 268 Here capitals are used in the in-text reference to the tables and figures. Further up a lower case was used to refer tot he appendices. Be consistent. With AMA/Vancouver these will be lower case.
238 The lack of diversity of participants is acknowledged. Could you comment on any efforts that were made to reach larger numbers and demographically diverse clinicians?
526 The lack of diversity is acknowledged and recommendations made to examine the efficacy of virtual delivery on diverse populations. Would you perhaps make mention of Indigenous and refugee populations here?
Minor edits:
Throughout - delete APA style in-text citations
56 Delete 'been'
93 delete 'set'
208 released 2020 - delete not needed
247 Full stop missing
253 Full stop missing
274 & 294 ry. ● indicates outlier. - would a semicolon be better than a full stop here?
322 'it was' twice - perhaps add a comma
337 avoiding avoidance - can this be punctuated, even if it is a direct transcript, a comma perhaps
448 spelling 'an' should be 'a'
Bibliography - formatting inconsistencies. Use commas to separate author names.

Author Response
Thank you very much for reviewing our manuscript. We greatly appreciate your time, suggestions and effort. All replies to the Reviewer's comments are in green font below.
General comments:
There are several spelling and punctuation errors. Please see comments in the attached file.
Thank you very much for catching these errors. We have made all of the necessary corrections.
The formatting of in-text citations and the bibliography needs attention.
The formatting of the in-text citations and bibliography have been fixed. Thank you for addressing these issues.
Can you comment on any differences in the perceived utility and safety of this approach with trauma-affected populations compared to other mental health difficulties not specifically attributed to traumatic experiences?
The use of digital mental health interventions is typically more well supported in psychiatric populations who's symptoms are not specifically attributable to traumatic experiences. Mental health clinicians generally perceive that digitally delivered psychotherapies are safe to provide general psychiatric populations. For example, there is growing evidence supporting the utility an safety of internet-based Cognitive Behavioral Therapy when used with civilian populations (Andrews et al., 2018; Carlbring et al., 2018).
In comparison, many clinicians perceive the digital delivery of trauma-focused psychotherapies to have lower safety and utility compared to in-person delivery. Many clinicians worry that the provision of trauma-focused psychotherapies to trauma-affected populations may negatively impact patient safety, as undergoing trauma-focused psychotherapy may increase the risk of experiencing symptom exacerbation or psychological distress. Such challenges could lead to increased treatment dropout rates (Larsen et al., 2016, Rozek et al., 2022, Burger et al., 2023). While evidence supporting the use of digitally delivered trauma-focused psychotherapies in trauma-affected populations exists (Burback et al., 2024), further research, such as the current study, is needed to change existing clinician perceptions regarding the use of digitally delivered trauma therapies.
Andrews G, Basu A, Cuijpers P, et al. Computer therapy for the anxiety and depression disorders is effective, acceptable and practical health care: An updated meta-analysis. J Anxiety Disord. 2018;55:70-78. doi: 10.1016/j.janxdis.2018.01.001.
Carlbring P, Andersson G, Cuijpers P, Riper H, Hedman-Lagerlöf E. Internet-based vs. face-to-face cognitive behavior therapy for psychiatric and somatic disorders: an updated systematic review and meta-analysis. Cogn Behav Ther. 2018;47(1):1-18. doi: 10.1080/16506073.2017.1401115.
Larsen SE, Wiltsey Stirman S, Smith BN, Resick PA. Symptom exacerbations in trauma-focused treatments: Associations with treatment outcome and non-completion. Behav Res Ther. 2016;77:68-77. doi: 10.1016/j.brat.2015.12.009.
Rozek DC, Baker SN, Rugo KF, Steigerwald VL, Sippel LM, Holliday R, et al. Addressing co-occurring suicidal thoughts and behaviors and posttraumatic stress disorder in evidence-based psychotherapies for adults: A systematic review. J Trauma Stress. 2022;35(2):729-745. doi: 10.1002/jts.22774.
Burger SR, Hardy A, van der Linden T, van Zelst C, de Bont PA, van der Vleugel B, et al. The bumpy road of trauma-focused treatment: Posttraumatic stress disorder symptom exacerbation in people with psychosis. J Traumat Stress. 2023;36:299–309. doi: 10.1002/jts.22907.
Burback L, Yap S, Purdon SE, et al. Randomized controlled trial investigating web-based, therapist-delivered eye movement desensitization and reprocessing for adults with suicidal ideation. Front Psychiatry. 2024;15:1361086. doi: 10.3389/fpsyt.2024.1361086.
Specific comments:
Line 55 please define PSP (public safety personnel)
We have defined PSP.
Lines 131-137 could this be summarised in a table, and can you explain how you selected participants for interviews/focus groups?
Thank you for your suggestion. We believe that this information regarding how the surveys were created is important and warrants being in the main text of the manuscript instead of a table.
When initially contacted, participants were given an opportunity to indicate their preference to be a part of an interview or focus group. Participants were part of an interview or focus group based on this response.
Line 268 Here capitals are used in the in-text reference to the tables and figures. Further up a lower case was used to refer to the appendices. Be consistent. With AMA/Vancouver these will be lower case.
Thank you for noting these inconsistencies. All in-text references to tables, figures, and appendices have been changed to lowercase.
238 The lack of diversity of participants is acknowledged. Could you comment on any efforts that were made to reach larger numbers and demographically diverse clinicians?
Thank you for this very thoughtful question. We had originally planned to only recruit clinicians living and working in Alberta. We quickly recognized that limiting our recruitment field to Alberta only was very restrictive. We therefore expanded our recruitment across Canada, with the hopes that this would allow us to get in contact with a more diverse group of clinicians. This strategy was unfortunately minimally effective, with few clinicians responding to our recruitment materials before recruitment was ceased due to reasons of feasibility.
526 The lack of diversity is acknowledged and recommendations made to examine the efficacy of virtual delivery on diverse populations. Would you perhaps make mention of Indigenous and refugee populations here?
We thank the reviewer for bringing this important suggestion to our attention. We have added “cultural background” to this part of the manuscript [page 13], to reflect how certain populations, such as Indigenous and refugee populations, may have moderating factors that could potentially impact their acceptance of digitally delivered care.
Minor edits:
Throughout - delete APA style in-text citations
Done.
56 Delete 'been'
Done.
93 delete 'set'
Done.
208 released 2020 - delete not needed
Done.
247 Full stop missing
Added.
253 Full stop missing
Added.
274 & 294 ry. ● indicates outlier. - would a semicolon be better than a full stop here?
We believe a full stop is best as this is the end of the caption for figures 1 and 2.
322 'it was' twice - perhaps add a comma
Added.
337 avoiding avoidance - can this be punctuated, even if it is a direct transcript, a comma perhaps
Added.
448 spelling 'an' should be 'a'
Changed.
Bibliography - formatting inconsistencies. Use commas to separate author names.
All references have been updated and changed to fit with journal requirements.
Reviewer 2 Report
Comments and Suggestions for Authors
Dear Authors:
First of all, I would like to congratulate you on this great research. It is a contribution to the study of treatments carried out through virtual methods.
There are some elements that, in my opinion, could be improved:
1. Although the introduction is well prepared, it would be prudent to incorporate other studies with other populations to clarify that this is one of the few studies on this type of treatment.
2. The work is very rigorous in terms of presenting the results. However, the study's small sample size remains a great weakness.
3. I think it would be prudent to add a couple of questions for each of the instruments used in the methodology.
4. In the discussion, I think it would be appropriate to discuss how these results could impact the forms of treatment, the training of health professionals, understanding that virtual therapy is already a supported methodology, and finally, how this study could affect public policies.
Finally, congratulations. This excellent study should be published so that the scientific community can learn from it.
Author Response
Thank you very much for reviewing our manuscript. We greatly appreciate your time, suggestions and efforts to strengthen the manuscript. Our replies to the Reviewer's comments are in green font below.
- Although the introduction is well prepared, it would be prudent to incorporate other studies with other populations to clarify that this is one of the few studies on this type of treatment.
Thank you for this suggestion. We agree with the reviewer that incorporating other studies would aid in strengthening our rationale for conducting this study. We have therefore cited two studies in the following passage to indicate that much of the existing literature has been focused on investigating the use of internet-based cognitive behavioral therapy in civilian populations. Please see the edited passage below.
“Much of the extant literature has also primarily focused on DMHI generally, such as the use of internet-based Cognitive Behavioural Therapy in civilian populations [17,18], with little focus on how the shift to digital delivery affected PPTE-focused treatments specifically. These challenges may contribute to the low acceptability of DMHI among clinicians [19], even with the knowledge that DMHI may provide many unique benefits.” [page 2, lines 75-79]
- Andrews G, Basu A, Cuijpers P, et al. Computer therapy for the anxiety and depression disorders is effective, acceptable and practical health care: An updated meta-analysis. J Anxiety Disord. 2018;55:70-78. doi: 10.1016/j.janxdis.2018.01.001.
- Carlbring P, Andersson G, Cuijpers P, Riper H, Hedman-Lagerlöf E. Internet-based vs. face-to-face cognitive behavior therapy for psychiatric and somatic disorders: an updated systematic review and meta-analysis. Cogn Behav Ther. 2018;47(1):1-18. doi: 10.1080/16506073.2017.1401115.
- The work is very rigorous in terms of presenting the results. However, the study's small sample size remains a great weakness.
Thank you for addressing this. We recognize that this is a limitation of our study and acknowledge this in the limitations section (pages 12-13).
- I think it would be prudent to add a couple of questions for each of the instruments used in the methodology.
Thank you very much for this helpful suggestion. We had considered doing so, however, felt it best to include all instruments used in this study in full as appendices (appendices 1-3), so as to (1) make the methodology and tools used as easy for readers to understand, (2) minimize potential misunderstandings of the tools that might arise if readers thought the sample questions were the full complement of the questions, and (3) limit redundancies.
- In the discussion, I think it would be appropriate to discuss how these results could impact the forms of treatment, the training of health professionals, understanding that virtual therapy is already a supported methodology, and finally, how this study could affect public policies.
Thank you for this very insightful comment. While we recognize the importance of these topics, the limited sample size of our study makes it difficult to speak directly on these topics based on the findings of this study. We do explore these key issues further in a separate publication, which included a larger sample size (Yap et al., 2024).
Yap S, Allen RR, Aquin CR, Bright KS, Brown MRG, Burback L, Winkler O, Jones C, Hayward J, Wells K, Vermetten E, Greenshaw AJ, Bremault-Phillips S. Current and Future Implementation of Digitally Delivered Psychotherapies: An Exploratory Mixed-Methods Investigation of Client, Clinician, and Community Partner Perspectives. Healthcare (Basel). 2024;12(19):1971. doi: 10.3390/healthcare12191971.